# Assessing species biomass contributions in microbial communities via metaproteomics

Manuel Kleiner [1,2], Erin Thorson [1], Christine E. Sharp[1], Xiaoli Dong[1], Dan Liu[1], Carmen Li[3] & Marc Strous [1]

Microbial community structure can be analyzed by quantifying cell numbers or by quantifying biomass for individual populations. Methods for quantifying cell numbers are already available (e.g., fluorescence in situ hybridization, 16S rRNA gene amplicon sequencing), yet high-throughput methods for assessing community structure in terms of biomass are lacking. Here we present metaproteomics-based methods for assessing microbial community structure using protein abundance as a measure for biomass contributions of individual populations. We optimize the accuracy and sensitivity of the method using artificially assembled microbial communities and show that it is less prone to some of the biases found in sequencing-based methods. We apply the method to communities from two different environments, microbial mats from two alkaline soda lakes, and saliva from multiple individuals. We show that assessment of species biomass contributions adds an important dimension to the analysis of microbial community structure.

[1] Department of Geoscience, University of Calgary, Calgary, AB, Canada T2N 1N4. [2] Department of Plant and Microbial Biology, North Carolina State University, Raleigh, NC 27695, USA. [3] Department of Biological Sciences, University of Calgary, Calgary, AB, Canada T2N 1N4. Manuel Kleiner and Erin Thorson contributed equally to this work. Correspondence and requests for materials should be addressed to M.K. (email: manuel_kleiner@ncsu.edu)

Microbial communities are ubiquitous in all environments on Earth that support life and they play crucial roles in global biogeochemical cycles, plant and animal health, and biotechnological processes[1]. One of the most basic and crucial parameters that microbial ecologists determine when studying these communities is their structure i.e., taxonomic composition and the relative abundances of the species in the community. Currently, all methods for assessing community structure provide a direct or indirect measure of cell numbers per taxon. For example, fluorescence in situ hybridization (FISH) provides direct cell counts[2], while metagenomics or 16S rRNA gene amplicon sequencing provide a more indirect measure of cell numbers as they essentially measure gene or genome copy numbers[3].

Cell numbers, however, are often not the best measure for a species' contribution to a community because microbial cells can differ by several orders of magnitude in biomass. For example, the unicellular eukaryote *Schizosaccharomyces pombe* has a cell volume and per cell proteinaceous biomass that is ~6000 fold higher than that of the bacterium *Mycoplasma pneumoniae*[4]. Therefore, the development of methods for the assessment of biomass contributions of community members is critical. Recently, FISH-based methods for the estimation of biovolume fractions of community members have been developed[5], however, these methods are limited to a few community members as a separate fluorescently-labeled probe is needed for each taxon that investigators want to analyze. Currently, there are no high-throughput methods available to estimate the biomass contribution of individual community members.

Metaproteomics is an umbrella term for methods for identifying and quantifying proteins in microbial communities[6] and may represent a suitable approach for assessing the structure of a microbial community based on species biomass contributions. Since proteins contribute a large amount of biomass in microbial cells e.g., 55% of *Escherichia coli* dry weight (BNID 104954, http://bionumbers.hms.harvard.edu/bionumber.aspx?id=104954)[7], proteinaceous biomass can be a good estimator of biomass contributions. Additionally, since proteins are the molecules that provide the biological activities to cells, metaproteomics may also provide estimates of activities. In recent years, several studies have been published, including some from our laboratory, which used the metaproteomic data to quantify biomass contributions of community members[8–10]. However, methods for biomass assessment with metaproteomics have not been thoroughly developed and validated, and several challenges and questions have not been addressed. The major challenge is the so-called protein inference problem of shotgun proteomic approaches[11]. In shotgun proteomics, which is the most widely used proteomic approach, proteins are identified by matching mass spectrometry derived peptide sequences to protein sequences. The protein inference problem describes the fact that often the same peptide sequence can match to multiple different proteins, which can lead to ambiguous protein identifications. This problem was originally noted for eukaryotes, which often have multiple, very similar isoforms of a protein[11]; however, the problem can be much more severe in metaproteomics because in metaproteomic analysis there are tens to hundreds of species that all have protein sequences sharing peptides with sequences from other species. The protein inference problem will thus lead to incorrect interpretations of taxonomic composition of metaproteomes[12] if not properly addressed. In fact, the protein inference problem is so pervasive that it has been advantageously used in metaproteomics for cross-strain and -species protein identification by using protein sequences from organisms closely related to the ones in the analyzed community[13,14]. Other challenges and questions include: How much mass

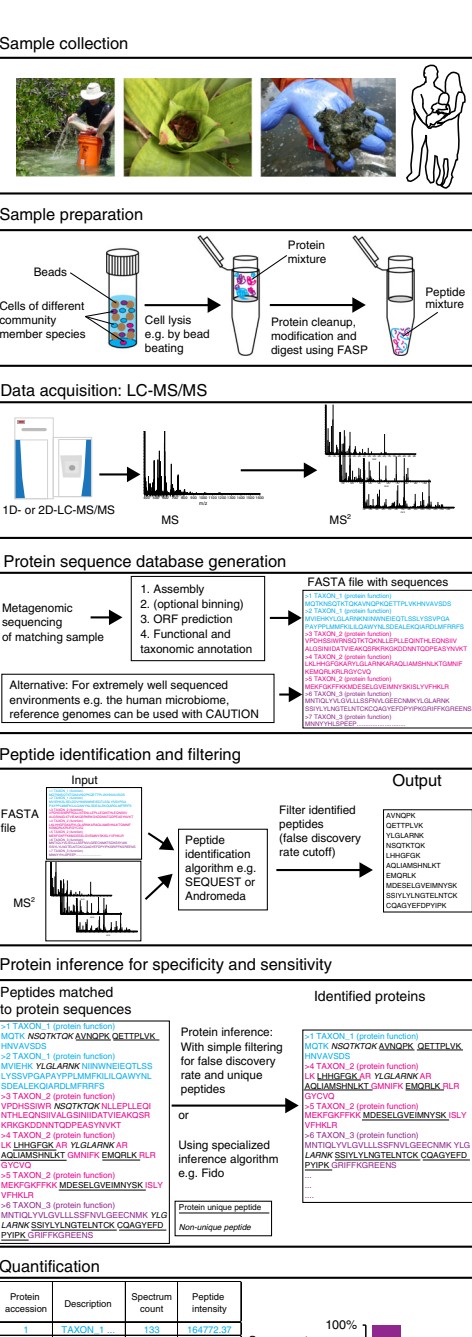

**Fig. 1** Workflow for assessing species biomass contributions using metaproteomics. The presented workflow can be adapted to different LC-MS/MS systems and computational tools. The critical considerations for achievement of high accuracy and sensitivity are as follows: (1) use of a protein sequence database derived from a metagenome of a sample matching to the metaproteomic samples; (2) selection and optimization of protein inference parameters with test datasets to achieve sufficient specificity; and (3) quantification of taxa using the sum of PSM counts or peptide intensities based on inferred proteins and not based on peptide inference

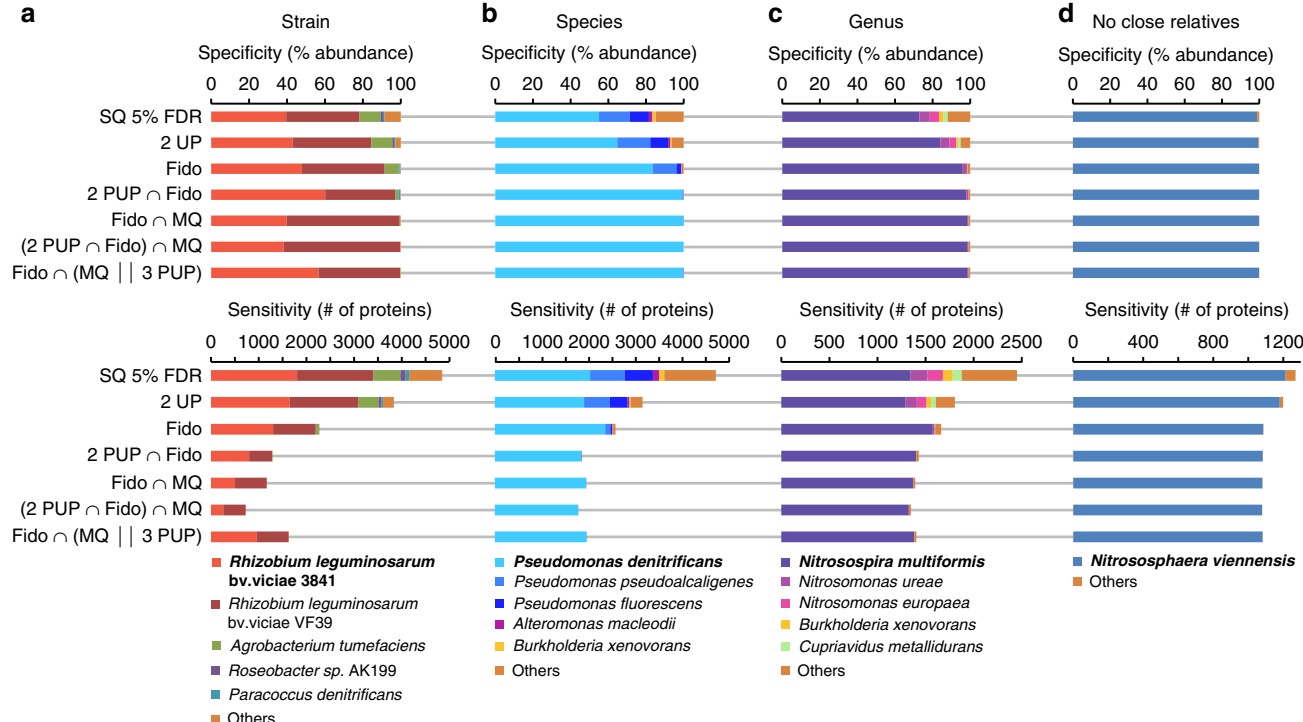

**Fig. 2** Specificity and sensitivity of protein identification with different protein inference strategies. The proteomics data for four pure cultures (**a** R. leg. bv. viciae 3841, **b** P. denitrificans, **c** N. multiformis, **d** N. viennensis) were analyzed in separate runs. For all cultures, proteins were identified with the same simulated metagenomic database containing 30 different species, which included other organisms related at the strain (**a**), species (**b**), or genus level (**c**). For **d**, no closely related organisms were included. Upper panels show specificity (% abundance of proteins attributed to each organism) and lower panels sensitivity (# of proteins identified). More stringent identification strategies improve specificity at the expense of sensitivity. Protein inference strategies: SQ 5% FDR; SEQUEST search filtered for 5% false discovery rate (FDR) using standard Target-Decoy strategy (implemented as Protein FDR Validator Node in Proteome Discoverer). 2 UP; same as previous, but only the subset of proteins identified with at least two unique peptides (UP). Fido; SEQUEST-Fido results filtered at 5% FDR based on protein q-value. 2 PUP ∩ Fido; same as previous, but only the subset of proteins identified with at least two protein unique peptides (PUP). Fido ∩ MQ; Only proteins identified both by Sequest-Fido (FDR 5%) and MaxQuant (1% protein FDR, at least 2 razor + unique peptides). (2 PUP ∩ SQ Fido) ∩ MQ; Same as previous, but for SEQUEST-Fido only the subset of proteins identified with at least two protein unique peptides. Fido ∩ (MQ ǁ 3 PUP); Only proteins that were identified both by SEQUEST-Fido and MaxQuant. Additionally, SEQUEST-Fido identified proteins were retained even if they were not identified by MaxQuant if they had at least three protein unique peptides

spectrometric data are needed to accurately quantify species in a community? And how do potentially incomplete protein sequence databases for protein identification affect the outcome of the quantification?

Here we address these challenges and questions to develop a simple and robust metaproteomics-based workflow for assessing species biomass contributions in microbial communities. Furthermore, we provide a large data set of metaproteomic, metagenomic and 16 S rRNA gene amplicon data from three types of artificial microbial communities for future method development and testing.

## Results

**Considerations for the basic workflow**. Overall, our method for species biomass assessment is similar to a basic workflow for metaproteomic protein identification and label-free quantification (Fig. 1). However, in contrast to protein and function-focused metaproteomics, the label-free quantification data (spectral counts or peptide intensities) are not summed for individual proteins, but rather for individual species or higher level taxonomic groups. Importantly, the quantification data are summed based on the taxonomic assignment of inferred proteins and not based on the taxonomic assignment of peptide identifications because, as mentioned above, peptides are frequently associated with multiple proteins from different taxa. Additionally, we

assume that a well annotated protein sequence database, which matches the studied environment as closely as possible, is used. This database could either be based on metagenomes derived from samples that match the metaproteomic samples or for well-studied environments, such as the human microbiome, a comprehensive, non-redundant set of sequences from public databases.

For this study, we used the Proteome Discoverer software (version 2.0, Thermo Scientific) and MaxQuant for protein identification, inference and quantification[15]. However, the methods discussed here are not platform dependent and can be implemented on many other platforms using the mock community data that we provide in this study for optimization.

**Achieving high specificity with minimal losses in sensitivity**. Before starting the actual species quantification, we first addressed the above mentioned protein inference problem. For this, we used proteomes from pure-culture organisms and simulated metagenomic databases to test what kind of protein inference parameters can be used to eliminate unwanted cross-strain and -species protein identifications (specificity), while still identifying a large number of proteins for quantification (sensitivity) (Fig. 2). We tested a variety of protein inference methods for the following four scenarios: the simulated metagenomic database contained the protein sequences of the analyzed organism and the sequences

of (a) a very closely related strain from the same species, (b) several closely related species from the same genus, (c) several related species from closely related genera, and (d) no other representative from the same domain (analyzed organism for (d) is an archaeon).

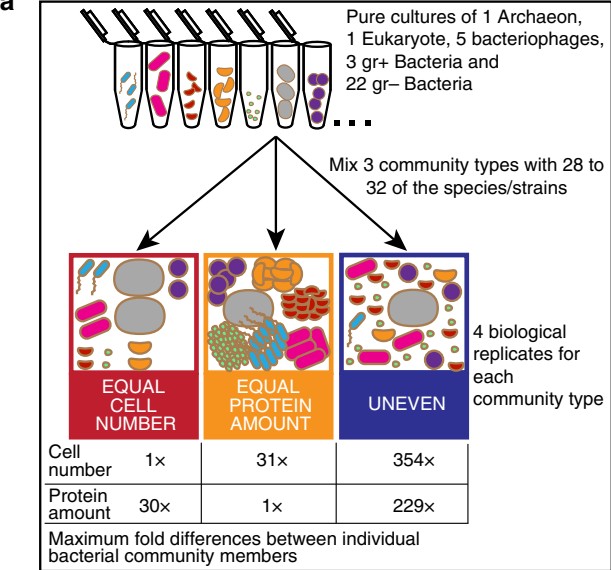

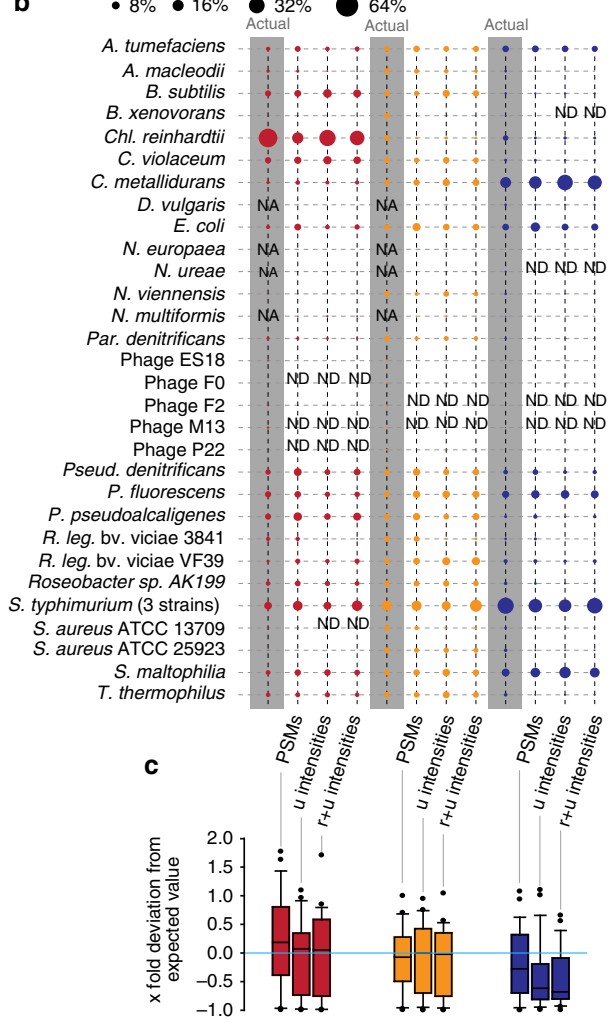

Commonly used protein inference filters that filter protein identifications simply for a false discovery rate (FDR) of 5% based on target-decoy database searches (SQ 5% FDR) fail to identify proteins from the analyzed organisms with high specificity for all scenarios except scenario (d) (Fig. 2). The same remains true when another commonly used criterion of requiring two unique peptides (2 UP) is added. Here, it is important to note that different protein identification platforms implement "unique peptides" differently. While, for example, a "unique peptide" in Proteome Discoverer and MaxQuant refers to a peptide that is unique to a group of highly similar protein sequences (protein group), it can also refer to a peptide that is unique to a single protein sequence in other protein identification platforms. This shows that to obtain high specificity on a taxonomic level protein inference has to be done differently from common practices.

We tested five additional protein inference and filtering strategies (Fig. 2) and found that there are multiple strategies that result in high specificity down to the species level i.e., removing almost all cross-species protein identifications, while at the same time maintaining a high sensitivity i.e., the number of identified proteins for the target organism is only slightly reduced as compared to the less specific approaches (Fig. 2). As expected, the approaches tested were unable to resolve cross-strain protein identifications in scenario (a) because protein sequences from the two strains were nearly identical in many cases. This suggests that it might be beneficial to remove highly similar sequences by sequence clustering when creating metaproteomic databases. Such a clustering would reduce database size, redundancy and the number of ambiguous strain level protein identifications, thus providing clearer species-level identifications.

Going forward, we used two protein inference strategies for this study. The first strategy relies on the SEQUEST algorithm for peptide identification and the Fido method for protein inference[16] (2 PUP ∩ Fido). Fido is available as a standalone program (https://noble.gs.washington.edu/proj/fido/) and as an advanced

**Fig. 3** Quantification of mock community using three proteomic quantification methods. **a** Illustration of mock community construction. 32 species and strains were used for the construction of three distinct community types. **b** Comparison of three proteomic quantification methods with the actual protein input amounts. Averages of four biological replicates per community are shown (full data in Supplementary Data 1). The data from two 460 min long 1D-LC-MS/MS runs were used per biological replicate. Three different quantification methods were used including sum of peptide-spectrum matches (PSMs), sum of peptide ion intensities using only unique peptides (u intensities), and sum of peptide ion intensities using razor and unique peptides (r + u intensities) as implemented in MaxQuant. The bacteriophages were mixed at a 1:10 ratio into the "equal protein amount" communities. On the basis of metagenomic sequencing, we found that the *B. xenovorans* culture was contaminated with *S. epidermidis* and thus the input protein amounts and cell numbers for *B. xenovorans* were lower than calculated. We used three *Salmonella enterica* typhimurium strains in the mock communities that differed only in a small number of genes and thus were de facto indistinguishable on the proteomic and metagenomic level and thus the inputs for the three strains are reported as one species. The bubble plot was generated with the bubble.pl script[55]. **c** Box plots show the x-fold deviation of the amounts measured with the three proteomic quantification methods from the actual protein input amounts. The box indicates the 1st and 3rd quartile, the line indicates the median and the whiskers indicate the 10th and 90th percentile. Outliers are indicated as individual points. If measurement and input were equal, then all values would be exactly 0 (indicated by bright blue line). Zeros (species that were not detected i.e., 'ND' in **b**) were removed before plotting. NA: species not added to this mock community; ND: Not detected with this method

implementation with convolution trees in Proteome Discoverer (FidoCT)[17]. For this strategy, only proteins that are identified by FidoCT with an FDR of 5% and have at least two protein unique peptides, are considered. The second strategy (SQ Fido ∩ (MQ ‖ 3 PUP), only considers proteins as confidently inferred if they are identified by both FidoCT (FDR of 5%) and MaxQuant (FDR of 1%, at least one unique peptide). Additionally, proteins are considered as confidently inferred if they have at least three protein unique peptides in the FidoCT result even if not identified by MaxQuant.

We are confident that many more strategies can be devised with the pure culture proteome data and the simulated metagenomic database, which we provide through the PRIDE repository (PXD006118).

**Metaproteomics enables accurate species-level quantification.** We used three types of mock communities to test and validate the methods for quantifying species biomass contribution in microbial communities. The three communities were assembled using 32 species and strains of Archaea, Bacteria, Eukaryotes and Bacteriophages (Fig. 3a, Supplementary Tables 1–3). Some of the bacterial strains were very closely related, but still distinguishable at the protein and nucleotide sequence level. These included the *Rhizobium leguminosarum* and *Staphylococcus aureus* strains. The three *Salmonella enterica* serotype typhimurium strains, however, only differed by a few mutations or the presence of an additional plasmid. The UNEVEN mock community was designed to cover a large range of species abundances both at the level of cell number and proteinaceous biomass to test for the dynamic range and detection limits of the quantification methods (Fig. 3a). The EQUAL PROTEIN AMOUNT and EQUAL CELL NUMBER mock communities contained either the same amount of protein for all community members with varying cell numbers or the same number of cells for all members with varying amounts of protein. Since the bacteriophages yield very little protein even if high particle numbers are used we mixed them at a 10x lower ratio into the EQUAL PROTEIN AMOUNT community.

We tested three of the most commonly used label-free quantification methods for their accuracy in measuring proteinaceous biomass contributions of individual species (Fig. 3b, c). These methods included counting and summing of peptide-spectrum matches (PSMs), summing of peptide ion intensities using only unique peptides (u intensities), and summing of peptide ion intensities using razor and unique peptides as implemented in MaxQuant ($r + u$ intensities)[15]. The input for these quantification methods were two 8 h long 1D-LC-MS/MS runs per sample (see methods section).

All three methods produced a good representation of the diversity in the mock communities and detected almost all species. The only exceptions were some of the bacteriophages and N. ureae, which were mixed into the samples in low total protein amounts (Fig. 3b). As expected, it was impossible to distinguish the three *Salmonella enterica* strains and thus they are represented in Fig. 3b as one row. All three methods performed similarly well when comparing the protein input amounts for the communities with the actual measurements (Fig. 3c). In most cases, the values for the measured % divided by the input % centered on the expected value of 1, with the median values being very close to 1. Differences between the quantification methods became apparent only for the UNEVEN community. Both peptide-intensity-based methods deviated strongly from the expectation and underestimated the abundance for many species. The PSM-based method was more robust for estimating abundances for the UNEVEN community which is characterized by large differences in cell numbers and total protein amount between species.

**Metaproteomics is more accurate for biomass estimates than sequencing methods.** We subjected subsamples of the above described mock communities to shotgun metagenomic sequencing and 16S rRNA gene amplicon sequencing to test how well these commonly used methods for community structure assessment estimate the proteinaceous biomass and cell number of species in communities in comparison to the metaproteomic method presented here.

We sequenced 16S rRNA gene amplicons for four biological replicates of each community type, yielding an average of 5356 high-quality amplicon sequences per replicate (minimum 1686 and maximum 9986 sequences). The amplicon sequences were clustered into 21 operational taxonomic units (OTUs) using the MetaAmp pipeline (version 1.3)[18,19]. Four of these OTUs were identified as Illumina in-run cross contaminants from unrelated samples that were sequenced on the same lane. The remaining 17 OTUs were taxonomically classified by MetaAmp at the genus level. A species-level classification was not possible because of the limited information content of the amplicon sequences. This meant, for example, that there were three OTUs that were classified as *Pseudomonas*. Therefore, we had to assign the OTUs to their respective species using BLASTn against the NCBI nr database and the prior knowledge about the content of our mock communities. As expected, none of the bacteriophages were detected by amplicon sequencing due to the absence of a 16S rRNA gene in these phages (Fig. 4a). We also did not detect the Archaeon *N. viennensis*, the eukaryotic green algae *Chl. reinhardtii* and six of the bacterial species by amplicon sequencing. The primer pair that we used to generate the amplicons is optimized for the greatest possible coverage of the bacterial domain[20], therefore, it was not surprising that *N. viennensis* and *Chl. reinhardtii* were not detected, although we successfully amplified at least the chloroplast sequence of green algae using this primer pair in the past (data not shown). The failure to detect some of the bacteria in all replicates is harder to explain. We have successfully generated amplicons from pure cultures of *N. europaea*, *N. ureae*, and *N. multiformis* in the past with the primer pair used here (data not shown), thus we have to assume that these species were not detected due to their low abundance in the UNEVEN community samples or due to a primer bias, leading to preferential amplification of the other bacterial species. Such primer biases are a known problem for 16S rRNA gene amplicon sequencing[3,21]. For the *R. leg.* bv. viciae and *S. aureus* strains, the amplicon sequences did not distinguish between each of the two strains in the samples and thus only a minimum of one strain detection per species could be corroborated.

Metagenomic sequencing of three biological replicates of each community type yielded on average 33.5 M 75 bp reads (max. 37 M, min. 21 M). The same DNA was used for the metagenomic sequencing and the 16S rRNA gene sequencing, however, only three of the four available biological replicates were metagenome sequenced. For quantification, we mapped the metagenomic reads to the reference genomes and assembled bins of the mock community members and normalized to the respective genome sizes.

All, except for one, organisms in the mock samples were detected by shotgun metagenomics, even including the single-stranded DNA bacteriophage M13. As expected, the only organism not detected by shotgun metagenomics was the single-stranded RNA bacteriophage F2 because the DNA extraction and sequencing library preparation methods used effectively exclude RNA from being sequenced. Surprisingly, the metagenomic sequencing yielded only a small number of reads for the green algae *Chl. reinhardtii*, which was in no way representative of the input cell number for the mock communities

(Fig. 4a). *Chl. reinhardtii* was much better represented in the metaproteomic data. One potential explanation for the under-representation of *Chl. reinhardtii* in the sequencing data could be a bias of the DNA extraction method used. The bead beating method used for DNA extraction, however, was quite rigorous. The metagenomic data provided by far the best representation of

the bacteriophages in the samples, with the exception of the F2 phage, which was only detected in the metaproteomes.

Comparing all three methods, metaproteomics provided the most accurate estimates of proteinaceous biomass for each species in the samples (Fig. 4b, c). The average *x*-fold deviations of the measured abundances from the expected abundance based on

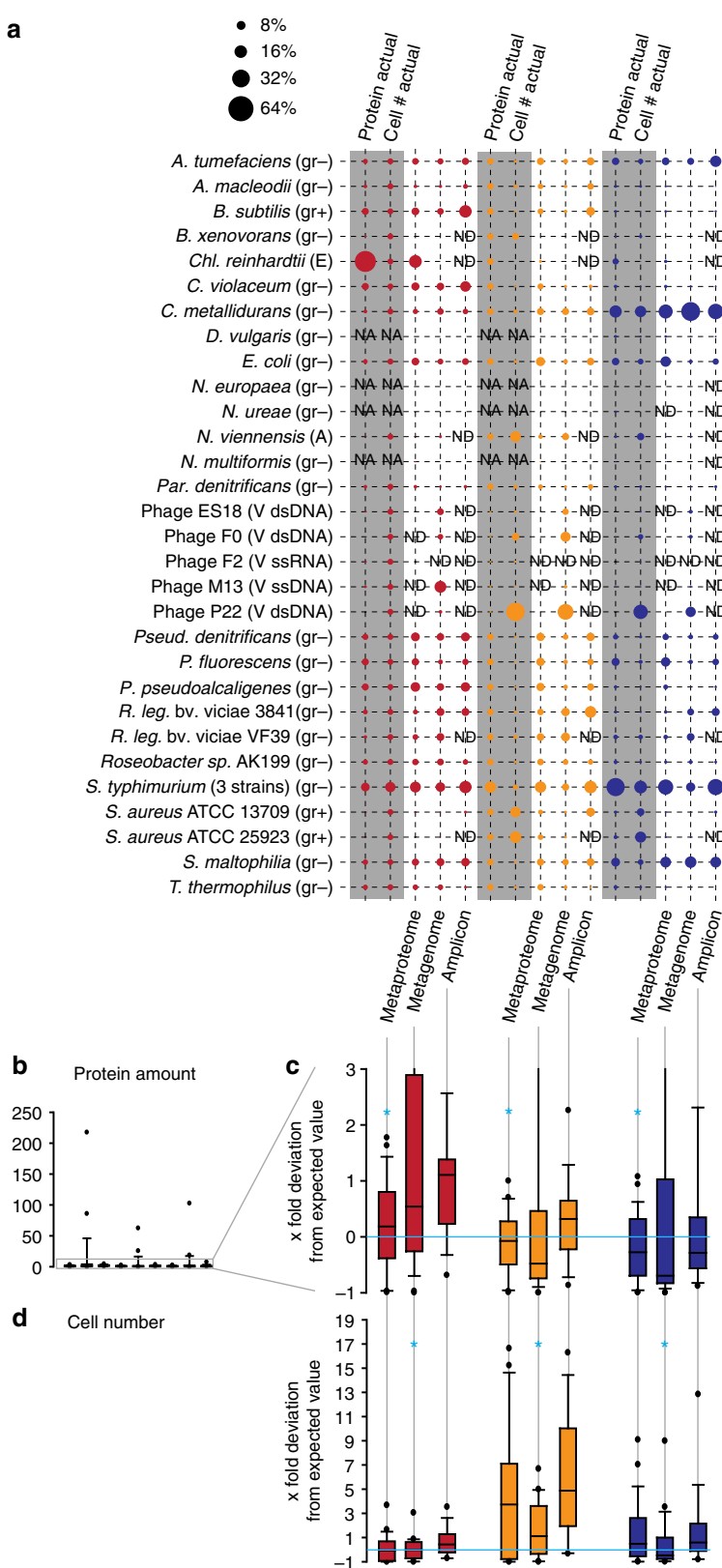

protein input were significantly lower for metaproteomics as compared to metagenomic and amplicon sequencing ($p$-value < 0.01, Supplementary Table 4). Both the metagenomic and the amplicon-based quantifications deviated from the actual values when it came to assessing proteinaceous biomass. Particularly, the metagenomic quantification produced some extreme outliers (Fig. 4b, Supplementary Table 4).

The relative species abundances provided by all three methods did not correlate well with the actual cell numbers in the samples (Fig. 4d). Overall, metagenomic sequencing provided the estimates closest to the actual cell number values, while the amplicon-based quantification deviated the most from the actual numbers. The average $x$-fold deviations of the measured abundances from the expected abundance based on cell input were significantly lower for metagenomics as compared to metaproteomics and amplicon sequencing ($p$-value < 0.01, $t$ test, Supplementary Table 4). The general overestimation of cell numbers by amplicon sequencing was in part due to the fact that the amplicon sequencing failed to detect many of the species in the mock communities driving up the relative abundances of the remaining ones.

Interestingly, the accuracy with which the three methods estimated the relative cell numbers in the mock communities depended very much on the range of species abundances in them. All three methods estimated the relative cell numbers quite well for the EQUAL CELL NUMBER community, but failed to estimate them well for the EQUAL PROTEIN AMOUNT and UNEVEN communities, which represent a large range of species abundances (Figs 3a and 4d). This is likely due to the more inaccurate quantification of low abundant strains/species that are close to the detection limit of the methods (see below and Fig. 5b).

**How much data is needed**. To test the impact of the number of spectra acquired on the detection limit and dynamic range of species proteinaceous biomass quantification, we ran five different LC-MS experimental setups for the four biological replicates of the UNEVEN mock community (Fig. 5, Supplementary Table 5). These setups provided varying numbers of $MS^2$ spectra for peptide identification. They included two basic 1D-LC-MS/MS approaches of 260 and 460 min run time. For each of these two approaches the amount of data was doubled by running technical replicates. The fifth approach was a 2D-LC-MS/MS experiment in which the sample was fractionated into 12 fractions using salt pulses on an SCX column followed by 120 min separations on a reverse phase column.

Each of the five approaches led to the detection of 27 out of the 30 distinguishable strains and species in the community when the biological replicates were combined. We observed some small differences between approaches in their detection sensitivity when looking at the data for individual biological replicates. While we detected 25–26 species/strains (average 25.25) in the single 260 min runs, we detected 26–27 (average 26.5) in the duplicate 460 min runs. From this follows that for the species

diversity and abundance distribution of the UNEVEN mock community a single 260 min (~130,000 $MS^2$ spectra) run provides a similar detection limit as compared to approaches that provide much more data (e.g., 2 × 460 min runs = ~ 390,000 $MS^2$ spectra). The detection limit for all five approaches was similar and, interestingly, differed by organism group. The Archaeon *N. viennensis*, the Eukaryote *Chl. reinhardtii* and all bacteria were detected with all five approaches. The bacterium *N. europaea* was mixed into the UNEVEN community with the lowest protein abundance of 0.08%, which suggests that at least for Bacteria the detection limit is below 0.08%. Three out of the five bacteriophages in the community were not detected by any of the approaches (Supplementary Table 6) even though they were mixed into the community at protein abundances higher than that of *N. europaea*, between 0.08 and 0.15%. This is surprising because these phages consist of only a few dominant proteins (e.g., capsid proteins), which should enhance their detectability. Currently, we do not have a good explanation for this result.

Surprisingly, all approaches had a similar accuracy in terms of quantifying species abundances (Fig. 5a). Our expectation was that an increased number of $MS^2$ spectra would increase the accuracy of the abundance estimates. Our data suggests that with a 260 min run we already reached saturation in terms of accuracy for the UNEVEN mock community type. Interestingly, all five approaches underestimated the abundances of species/strains that are present in the samples in low amounts (Fig. 5a). If low-abundance species (<0.5% in all approaches) are removed from the data set, resulting in 18 species remaining, then the deviation of the measurement from the actual protein input amount becomes much smaller (Fig. 5b, Supplementary Table 6). This suggests that, as with most other analytical methods, the accuracy of the measurement is lower for quantities close to the detection limit and thus the proteinaceous biomass estimates for low abundant species should be treated as less precise.

In summary, a single 260 min 1D-LC-MS/MS run on a QExactive Plus Mass Spectrometer provides enough data to detect most species in a community that contains 30 distinguishable species and features a range of proteinaceous biomass abundances of more than two orders of magnitude. The limit of detection can be slightly lowered using longer peptide separations and by increasing the amount of data generated per sample.

**Absolute biomass estimates with incomplete sequence databases**. One potential drawback of metaproteomics-based biomass quantification of species in a microbial community is that proteomic protein identification relies on the availability of a protein sequence database. Proteins can only be identified and quantified if protein sequences are present in the database that have a high similarity to the actual proteins in a sample. Analogous to the primer, bias-based exclusion or incorrect estimation of species abundances in 16S/18S rRNA gene amplicon sequencing[20,22], the incompleteness of the protein sequence database used for protein identification can lead to the exclusion or incorrect estimation of

**Fig. 4** Comparison of metaproteomic, shotgun metagenomic and 16S rRNA gene amplicon-based quantification of the mock communities. **a** The same mock communities as in Fig. 3 were used. For 16S rRNA gene amplicons and metaproteomes averages of four biological replicates per community type are shown. For the shotgun metagenomic data averages of three biological replicates are shown. For the metaproteomes the PSM-based quantification is shown (Fig. 3b). (gr − or gr + ) Gram-positive or -negative bacterium, (A) Archaeum, (E) Eukaryote, (V dsDNA, ssDNA or ssRNA) virus specifying nucleic acid type of genome. **b–d** Box plots show the $x$-fold deviation of the species abundance quantification with metaproteomics, metagenomics and 16S rRNA gene amplicon sequencing from the actual input amounts for protein (**b**) and (**c**), and cell number (**d**). **c** is an enlargement of the lower part of (**b**). If measured and input species abundance were equal, then all values would be exactly 0 (indicated by bright blue line). Zeros (species that were not detected i.e., 'ND' in **a**) were removed before plotting. For each community type and method the method with the significantly lowest $x$-fold deviation (p-value < 0.01, $t$ test) is indicated with a bright blue '*' (see Supplementary Table 4 for details on statistics). NA species not added to this mock community, ND not detected with this method

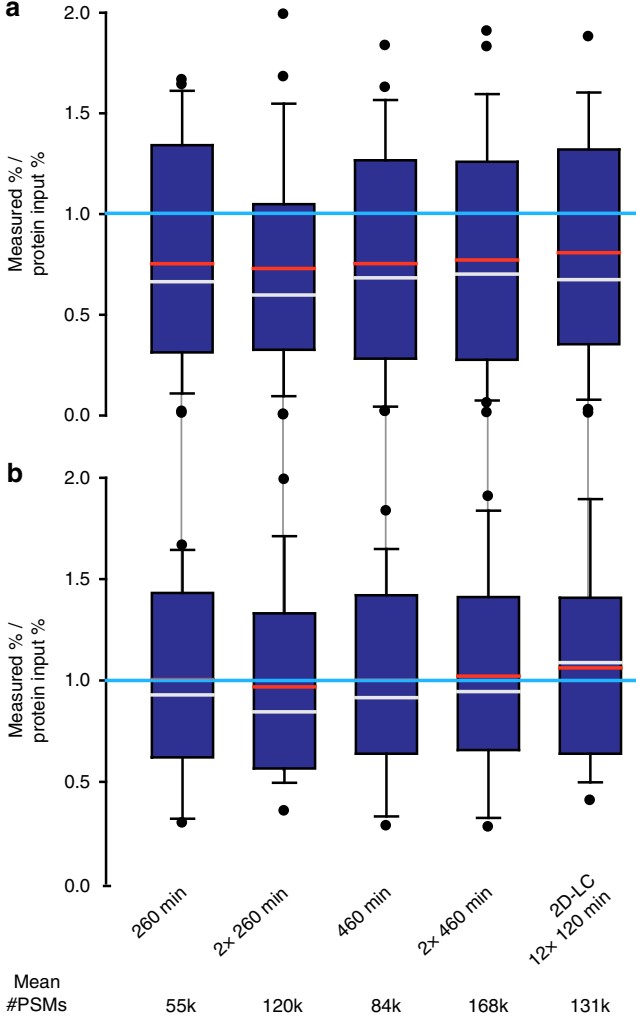

Mean
#PSMs

| 260 min | 2× 260 min | 460 min | 2× 460 min | 2D-LC 12× 120 min |
|---------|-----------|---------|-----------|-------------------|
| 55k | 120k | 84k | 168k | 131k |

**Fig. 5** Comparison of quantification accuracy for the UNEVEN mock community depending on LC method, gradient length and technical replication. Four biological replicates were run for each method and the number of peptide-spectrum matches (PSMs) was averaged per organism. The numbers below the plots give the average total number of PSMs generated for each of the methods. Box plots show the deviation of the amounts measured by summing of PSMs from the actual protein input amounts. The box indicates the 1st and 3rd quartile, the grey line indicates the median, the red line the average and the whiskers indicate the 10th and 90th percentile. If measurement and input were equal, then all values would be exactly 1 (indicated by bright blue line). The first four methods were 1D-LC-MS/MS runs of the given length. The '2×' indicates if technical replicates were run. The fifth method was 2D-LC-MS/MS runs with each of the 12 fractions measured for 120 min (detailed data for this Figure is in Supplementary Table 6). In **a**, the deviation values for all 27 detected strains and species are shown. In **b**, the deviation values are only shown for the 18 strains and species that had an abundance > 0.5% based on at least one of the methods

species abundances. However, the metaproteomic data in theory allows estimating how incomplete the sequence database used is based on the number of available mass spectra and the known proportion of how many of these mass spectra lead to PSMs in a search with a mock community for which all protein sequences are known. This should allow correction of the relative abundance estimates to absolute estimates.

To test the influence of database incompleteness on quantification results and if the error in abundance estimates resulting from

it can be corrected for, we used two sequence databases of varying incompleteness to quantify the species in the UNEVEN community. In the first incomplete database (INCOMPLETE1), the protein sequences for *Pseudomonas denitrificans*, *Pseudomonas fluorescens* and *Rhizobium leguminosarum* bv. viciae strain 3841 were removed leaving the sequences of the closely related species/ strains *Pseudomonas pseudoalcaligenes* and *Rhizobium leguminosarum* bv. viciae strain VF39 in the database. In the second incomplete database (INCOMPLETE2), the remaining *Pseudomonas* and *Rhizobium* sequences as well as the *Salmonella enterica* typhimurium LT2 sequences were removed.

As expected, the number of detected organisms dropped for the quantification with the incomplete sequence databases (Fig. 6a). In the quantification with the INCOMPLETE1 database, the number of PSMs for the remaining *R. leg.* VF39 and *P. pseudoalcaligenes* increased and thus their relative abundance. This increase in PSM number is due to the fact that in the absence of the protein sequences of the correct species/strain some of the MS$^2$ spectra match to peptides from closely related species/ strains. As expected, for the very closely related *R. leguminosarum* strains, a larger fraction of PSMs shifted from one strain to the other as compared to the *Pseudomonas* species for which only a smaller fraction of PSMs shifted over. The PSM number for most remaining organisms remained very similar across the database completeness range with the exception of *E. coli*, which obtained a large number of additional PSMs from the closely related *S. enterica* in the quantification based on the INCOMPLETE2 database. As expected, the drop in the total number of PSMs led to an increase of relative organism abundance when more protein sequences were removed from the database (Fig. 6a, b). We corrected these relative biomass estimates by calculating the number of PSMs lost due to database incompleteness based on the known proportion of MS$^2$ spectra to PSMs in the quantification with the complete database. The corrected relative abundance estimates for the quantification with the INCOMPLETE2 database were in most cases very similar to the quantification with the complete database (Fig. 6b). Therefore, the proteinaceous biomass abundances adjusted for database incompleteness can be used as an approximation of absolute proteinaceous biomass abundances.

**Case studies**. To demonstrate the power and application of the metaproteomics-based methods for assessing species biomass contributions in microbial communities, we applied the methods developed here to microbial communities from two widely different environments.

For the first application example, we generated metaproteomic data, as well as the 16S rRNA gene amplicon data from two phototrophic biomats from soda lakes in the Canadian Rocky Mountains (Fig. 7a). We summarized organism abundances at the phylum level. Even on this high taxonomic level, major differences between the lakes and the two approaches become apparent. While the 16S rRNA gene amplicon data suggests that the lakes were rather similar in community structure on the phylum level, the metaproteomes painted a very different picture. The metaproteomes indicate that the major phototrophs between the lakes were different. Lake 1 was dominated by Cyanobacteria, whereas lake 2 was dominated by green algae. Additionally, we detected dsDNA viruses in lake 2, which despite the fact that they contribute only a small amount of proteinaceous biomass could play an important ecological role. Interestingly, some bacterial groups that made up a significant amount of the 16S rRNA gene amplicons (e.g., bacteroidetes/Chlorobi group) contributed only a minor amount based on the metaproteomic data. Since the cell lysis method used for both approaches was identical, an

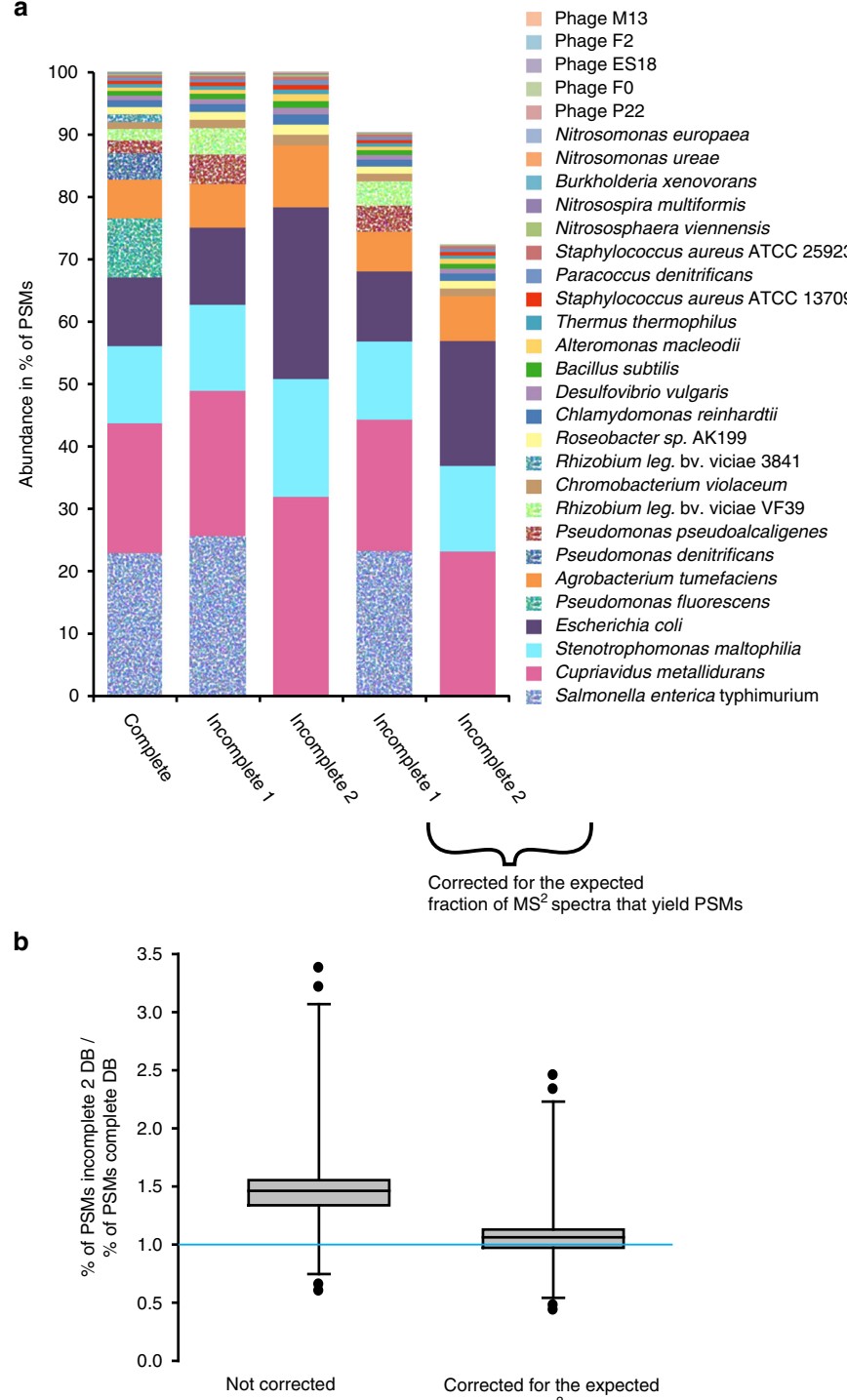

**Fig. 6** Effect of using incomplete protein sequence databases on quantification results. Species in the UNEVEN community were quantified using the complete protein sequence database containing the protein sequences of all species in the community for protein identification and two sequence databases of varying incompleteness. In the first incomplete database (INCOMPLETE1), the protein sequences for *Pseudomonas denitrificans*, *Pseudomonas fluorescens* and *Rhizobium leguminosarum* bv. viciae (strain 3841) were removed leaving the sequences of the closely related species/strains *Pseudomonas pseudoalcaligenes* and *Rhizobium leguminosarum* bv. viciae (strain VF39) in the database. In the second incomplete database (INCOMPLETE2) the remaining *Pseudomonas* and *Rhizobium* sequences as well as the *Salmonella enterica* typhimurium LT2 sequences were removed. In **a**, the average quantification results for the four UNEVEN community biological replicates are shown. For the 4th and 5th bar the quantification results were corrected by considering the percentage of PSMs lost due to database incompleteness, i.e., based on searches with the complete database the expected fraction of MS$^2$ spectra that yielded PSMs was known and thus we could calculate the difference between the expected number of PSMs and actual number of PSMs in the quantification with the incomplete databases. In **b**, the comparison of the quantification results with the complete and the INCOMPLETE2 databases are shown before and after correction of the INCOMPLETE2 quantification results. If the quantification results were in perfect agreement then all values would be 1 (indicated by the bright blue line). The box indicates the 1st and 3rd quartile, the line indicates the median and the whiskers indicate the 10th and 90th percentile

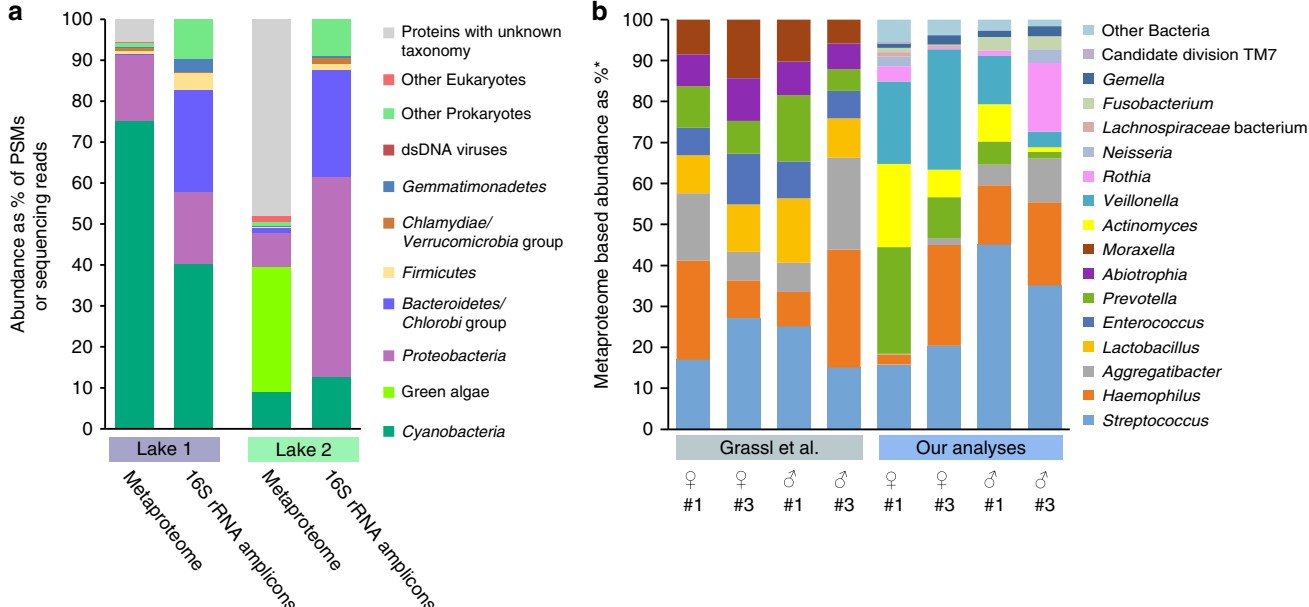

**Fig. 7** Two application examples using metaproteomics of estimating species biomass in communities. **a** Comparison of phylum level quantification of two soda lake biomats using metaproteomics and 16S rRNA gene amplicon sequencing. **b** Re-analysis of a published saliva metaproteome data set by Grassl et al.[9] using our method and comparison with the original analysis. *For the Grassl et al. analyses the abundances are given in % of the summed MS intensities of the 10 most abundant peptides per genus across all samples. For our analyses the abundances are given as % of all PSMs from proteins inferred by FidoCT with an FDR of 5% and at least 2 protein unique peptides

extraction bias is unlikely, suggesting that other causes such as a primer bias or relic DNA[23,24] may be responsible for the discrepancy.

For the second application example, we re-analyzed a recently published saliva metaproteome that provided extensive insights into the diurnal and inter-individual variation of the oral microbiome[9]. Grassl et al.[9] provided two independent datasets in their study on the presence and abundance of specific taxa in the oral microbiomes. The first data set (Fig. 4 in the original publication) provides presence/absence patterns of taxa based on unique peptide matches and cultivation results. The second data set provides quantification of taxa based on peptides identified by metaproteomics, however, without the specificity increasing step of protein inference (Fig. 6c in the original publication). Our results from the re-analysis of the proteomic data corresponded well with the taxonomic presence and absence patterns inferred by Grassl et al.[9]. However, our metaproteomic quantification of the data showed very different abundance and presence profiles for bacterial genera, as compared to the original metaproteomic analysis. We observed a much larger inter-individual variation for organism abundances (Fig. 7b). Additionally, several genera that were detected in the original analysis to be abundant in the samples were not detected at all or in much lower abundances (e.g., *Enterococcus* and *Abiotropha*), while other genera were much higher in abundance (e.g., *Veillonella*, *Actinomyces* and *Rothia*) (Fig. 7b). Grassl et al.[9] acknowledged in their study that the quantification method they used could come "at the disadvantage that peptides shared by two genera could lead to an overestimation of the taxon's abundance." Our analyses suggest that non-unique matching of peptides between genera indeed led to the skew in the original quantification data. For example, *Enterococcus* and *Abiotropha* share many peptides with *Streptococcus*, however, only streptococcal proteins could be inferred confidently to be present in the samples. This demonstrates that using validated, highly specific protein inference criteria for metaproteomic-based species quantification is crucial and that peptide identification without subsequent protein inference is not sufficient to achieve high enough specificity for quantification.

## Discussion

The metaproteomics-based biomass assessment approach that we demonstrate here is not limited to a specific set of computational tools and parameters. To make the approach as broadly applicable as possible, we have chosen to use a route that should make it possible to transfer the approach to many other platforms, including some recently developed software pipelines addressing the protein inference problem in metaproteomics such as The MetaProteomeAnalyzer[25] and Prophane[26]. In this manuscript, we highlight the most crucial considerations for developing concrete methods for metaproteomics-based biomass assessment (e.g., protein inference specificity) and supply a comprehensive data set to transfer this approach to other computational or experimental platforms for proteomics. The provided pure-culture-derived proteomes (PXD006118), for example, will allow investigators to determine parameters to achieve sufficient protein inference specificity, while the different mock community proteomes (PXD006118) will allow assessing parameters based on quantification accuracy and number of detected species.

As we demonstrate here, metaproteomics-based biomass assessment is a powerful approach that allows accurate quantification of the proteinaceous biomass of a large number of taxa in a community all at once. This approach augments existing high-throughput approaches for determining community structure based on DNA sequencing, in that it provides an additional, independent measure of community structure. Our case study on soda lake biomass nicely illustrates that sequencing-based methods and metaproteomics can provide very different pictures of a community. An added benefit of using metaproteomes in addition to sequencing-based methods for community structure analyses is that the proteomic information will also provide insights into which metabolic and physiological functions are expressed and play a major role in the community.

Recently, there has been a recurring interest in more quantitative methods for microbial ecology for the absolute quantification of community structure (e.g., cell counts per volume)[27]. Metaproteomics-based abundance estimates can be put into an absolute context by simple assays, for example, by measuring total protein content of a specified sample volume, wet weight or dry weight. The relative proteinaceous biomass abundances of community members can then be converted to absolute values after considering necessary corrections for database incompleteness (see Results).

There are several questions that go beyond the scope of this study that should be addressed in the future. First, is proteinaceous biomass an accurate representation of the total biomass of a species? We would argue that, in many cases, proteinaceous biomass is a good estimate of total biomass, because it has been shown for a variety of bacteria that the ratio of protein to total cell dry weight is relatively constant even for different growth states[28–30]. However, as always, we expect exceptions, where proteinaceous biomass is not a good predictor of total biomass, which would, for example, be the case of microorganisms that store large amounts of carbon in form of polyhydroxyalkanoates or glycogen. Second, a likely much more difficult question to answer is, can proteinaceous biomass of a community member be used as an approximation of the biological activity of that community member, and if so under what circumstances? This question can potentially be addressed in the future by combining the metaproteomics-based biomass assessment approach with methods that allow determination of species-specific activities based on incorporation of stable isotopes on the single-cell level such as NanoSIMS[31] and Raman microspectroscopy[32] or community-level by metaproteomics using Protein-SIP[33].

## Methods

**Assembly of mock communities.** Cultures of 32 Archaea, Bacteria, Eukaryotes and Bacteriophages (Supplementary Tables 1 and 7) were donated to us by very kind colleagues. The cells were washed using phosphate buffered saline, pH 7.4 (Sigma-Aldrich) to remove the cultivation medium. The cell counts of washed cells were determined by microscopy using a Neubauer improved counting chamber. Cells were aliquoted and pelleted by centrifugation at $21,000\,g$ for 5 min to create cell aliquots with known cell number. Bacteriophages were purified by filtration and polyethylene glycol (PEG) precipitation as described in Kleiner et al.[34]. Phage titers were determined as particle forming units (PFUs) per ml using the soft-agar overlay method[35]. Liquid aliquots with known titer were made for all phages. Cell pellets and phage aliquots were stored at −80°C.

We quantified the protein content of cell and phage aliquots for each strain using duplicate aliquots. For this, 300–600 μl SDT-lysis buffer (4% (w/v) sodium dodecyl sulfate (SDS), 100 mM Tris-HCl pH 7.6) were added to each pellet according to pellet size. The pellets in SDT-lysis buffer were vortexed and transferred to lysing matrix tubes (Matrix A, MP Biomedicals, Santa Ana, CA, USA) and lysed using a Bead Ruptor 24 (Omni International, https://www.omni-inc.com/) at $6\,m\,s^{-1}$ for 45 s. The samples were heated for 10 min to 95 °C and then centrifuged for 10 min at $21,000\,g$. Dilutions of each sample were prepared and sample protein amounts were quantified using the Pierce bicinchoninic acid (BCA) assay (Thermo Scientific Pierce).

We assembled three types of mock communities by resuspending the frozen cell pellets of each microorganism in 150 μl ultrapure water and then combining varying amounts of each organism. The structure of each mock community type is detailed in Supplementary Tables 1–3. Four biological replicates of each mock, community type were made and each replicate was divided into 20 aliquots. The aliquots were frozen at −80 °C until extraction. The UNEVEN mock community was designed to cover a large range of species abundances both on the level of cell number and proteinaceous biomass to test for the dynamic range and detection limits of the quantification methods (Fig. 3a). The EQUAL PROTEIN AMOUNT and EQUAL CELL NUMBER mock communities contained either the same amount of protein for all community members with varying cell numbers or the same number of cells for all members with varying amounts of protein. Since the bacteriophages yield very little protein even if high particle numbers are used, we mixed them at a 10× lower ratio into the EQUAL PROTEIN AMOUNT community.

**Sampling of soda lake biomats.** Benthic microbial mats were sampled from two soda lakes located on the Cariboo Plateau, British Columbia, in June 2014 for 16S rRNA gene amplicon sequencing and metaproteomics and in May 2015 for metagenomic sequencing. Lake 1 herein refers to Goodenough Lake (51°19'47.64″

N 121°38'28.90″W) and Lake 2 refers to Last Chance Lake (51°19'39.3″ N121° 37'59.4″W). Collected microbial mats from each lake were pooled and immediately placed on ice in the field and frozen at −80 ° C within two days of sampling for DNA extraction.

**DNA extraction.** For the mock community samples, DNA was extracted from one aliquot of each of the four biological replicates of each community type using the FastDNA Spin Kit (MP Biomedicals, Santa Ana, CA, USA) according to the manufacturer's protocol with small modifications. Following addition of CLS-TC to each aliquot, samples were homogenized in lysing matrix tubes (MP Biomedicals FastDNA Spin Kit, tube A) for 45 s at $6\,m\,s^{-1}$ using a Bead Ruptor 24 (OMNI). In addition, the DNA elution step was repeated twice. DNA concentrations were measured using a NanoDrop 2000 spectrophotometer (Thermo Scientific).

DNA was extracted from the 2014 and 2015 Lake 1 and Lake 2 samples using the FastDNA Extraction Kit for Soil (MP Biomedicals) with 10 min centrifugation times for the spin filter steps and an additional purification using 5.5 M guanidine thiocyanate as described in Sharp et al.[18].

**16S rRNA gene amplicon sequencing.** DNA from all mock community samples and the 2014 soda lake biomats from Lake 1 and Lake 2 was used for 16 S rRNA gene amplicon libraries preparation as described in Sharp et al.[18]. We used the S-D-Bact-0341-a-S-17 (also known as b341, 5′-TCGTCGGCAGCGTCA-GATGTGTATAAGAGACAGCCTACGGGAGGCAGCAG-3′)[36] and S-D-Bact-0785-a-A-21 (also known as Bakt_805 R, 5′-GTCTCGTGGGCTCGGA-GATGTGTATAAGAGACAGGACTACHVGGGTATCTAATCC-3′)[37] primers (primer sequences underlined) with added Illumina overhang adapters for the amplification of the HV regions 3–4, resulting in 427 bp amplicons (excluding the primers). Based on the evaluation by Klindworth et al.[20] this primer pair yields a large coverage of the domain Bacteria. Libraries were pooled and normalized for sequencing on the Illumina MiSeq Sequencer (San Diego, CA) using the 2 × 300 bp MiSeq Reagent Kit v3. The resulting amplicon sequences were analyzed with MetaAmp[18,19]. Operational taxonomic units (OTUs) were identified with a threshold of 97% sequence similarity.

**Metagenomic sequencing of mock communities.** Shotgun metagenomic sequencing (2 × 75 bp) of three replicates of each mock community type was performed using the Illumina NextSeq 500 sequencer. The NEBNext Ultra II DNA Library Prep Kit (New England Biolabs) was used for library preparation. Ten to nineteen million paired-end reads were generated for each sample. We confirmed the library content using PhyloFlash (https://github.com/HRGV/phyloFlash) and the quality of the data using FastQC (http://www.bioinformatics.babraham.ac.uk/projects/fastqc/). We used BBsplit from the BBmap package (version 35.85, http://sourceforge.net/projects/bbmap/) to map raw reads against the mock community reference genomes to quantify the read coverage for each organism. The reference genomes that were used are listed in Supplementary Table 8. Read mapping statistics for each reference genome were generated using BBsplit's default parameters and by setting the 'refstats' parameter. Relative read abundances for each organism were normalized to their genome sizes.

No reference genomes were available for *Roseobacter* sp. AK199 and *Chromobacterium violaceum* CV026 in public databases. Therefore, we generated genomes for these two strains from the metagenomes using an iterative assembly and binning strategy. All read files were trimmed for quality and adapters using BBduk from the BBmap package (http://sourceforge.net/projects/bbmap/). The trimmed reads for the UNEVEN samples were concatenated and assembled with metaSPAdes (version 3.8.1)[38]. The assembly quality was checked by running metaQUAST (version 4.1) with the mock community reference genome set[39]. Metawatt (version 3.5.2) was then used to create bins for AK199 and *C. violaceum* CV026 using default settings[40]. The bins were checked with metaQUAST to ensure that none of the included contigs aligned with any of the other reference genomes for the mock community. The trimmed reads from all samples were concatenated and BBmap was used to retrieve reads mapping to the AK199 and *C. violaceum* bins. SPAdes (version 3.8.1) was used to assemble the mapped reads for AK199 and *C. violaceum*[41]. The assembly quality was checked with metaQUAST, QUAST, and CheckM[42]. The AK199 genome was of sufficient quality after this assembly round. The *C. violaceum* assembly was further improved by two more rounds of read mapping and assembly. The AK199 and *C. violaceum* genomes were annotated using the RAST server[43] and annotated protein sequences were retrieved for the construction of the protein identification database.

**Soda lake biomat metagenomes.** DNA (250 ng) from the 2015 soda lake biomats from Lake 1 and Lake 2 was randomly sheared to a fragment size of approximately 300 bp using a S2 focused-ultrasonicator (Covaris, Woburn, MA). The fragmented DNA was then converted into an Illumina compatible sequencing library using the NEBNext Ultra DNA Library Prep Kit according to the vendor's standard protocol. This included a size selection step with SPRIselect magnetic beads and PCR enrichment (8 cycles) with NEBNext Multiplex Oligos for Illumina. The libraries were measured using qPCR and the Kapa Library Quant Kit for Illumina and then pooled in equal amounts for sequencing. A 1.8 pM solution was then sequenced on an Illumina NextSeq 500 sequencer using a 300 cycle (2 × 150 bp) high-output

sequencing kit as per the Illumina protocol in the Center for Health Genomics and Informatics in the Cumming School of Medicine, University of Calgary. All raw Illumina reads were passed through an in-house Illumina read quality control program that filters out known Illumina sequencing and library preparation artifacts. Specifically, all reads were removed that: (i) matched the spike-in PhiX sequence; (ii) were shorter than 30 bp after clipping off the partial primer, adapters, and the low-quality ranging at the ends; or (iii) were of low complexity. Reads that passed the quality control stage were assembled into contigs using MEGAHIT v1.0.3 with options "--k-list 51,77,99,127 --min-count 2 –min-contig-len 500"[44]. The assembled contigs were merged into scaffolds based on paired-end information using the SOAP v2.04 package[45]. The GapCloser v1.12 package was applied to further close the gaps between contigs in scaffolds. All the scaffolds longer than 500 bp after GapCloser post-processing were run through Prodigal v2.6.1 to identify coding sequences[46]. The coding sequences ( > = 60 aa) were annotated using DIAMOND[47] with options "-k 1 --seg no" to search against a protein sequence reference database generated by GenomeDatabase (https://sourceforge.net/projects/genomedatabase/) and the eggNOG database[48]. Best-hit matches were filtered by query coverage > = 70% and percent identity > = 30%. Taxonomic assignments for protein sequences were made on the basis of the filtered best-hit matches. The taxonomically annotated protein sequences were then used to generate the protein identification database, by combining them with protein sequences from several eukaryotic genomes and transcriptomes, which were chosen based on the results from a 18S rRNA amplicon library. CD-HIT was used to remove redundant sequences from the database using an identity threshold of 95%[49]. The cRAP protein sequence database (http://www.thegpm.org/crap/) containing protein sequences of common laboratory contaminants was appended to the database. The final database contained 4,171,024 protein sequences and is available from the PRIDE repository (PXD006343).

**Sample preparation for proteomics.** Samples were lysed in SDT-lysis buffer with 0.1 M DTT. SDT-lysis buffer was added in a 1:10 sample/buffer ratio to the sample pellets. The cells were disrupted in lysing matrix tubes A (MP Biomedicals) for 45 s at 6 m s$^{-1}$ using the OMNI Bead Ruptor 24 and subsequently incubated at 95 °C for 10 min followed by pelleting of debris for 5 min at 21,000 g. We prepared tryptic digests following the filter-aided sample preparation (FASP) protocol described by Wisniewski et al.[50]. In brief, 30 μl of the cleared lysate were mixed with 200 μl of UA solution (8 M urea in 0.1 M Tris/HCl pH 8.5) in a 10 kDa MWCO 500 μl centrifugal filter unit (VWR International) and centrifuged at 14,000 g for 40 min. 200 μl of UA solution were added again and centrifugal filter spun at 14,000 g for 40 min. 100 μl of IAA solution (0.05 M iodoacetamide in UA solution) were added to the filter and incubated at 22 °C for 20 min. The IAA solution was removed by centrifugation and the filter was washed three times by adding 100 μl of UA solution and then centrifuging. The buffer on the filter was then changed to ABC (50 mM Ammonium Bicarbonate), by washing the filter three times with 100 μl of ABC. 1 to 2 μg of MS grade trypsin (Thermo Scientific Pierce, Rockford, IL, USA) in 40 μl of ABC was added to the filter and the filters were incubated overnight in a wet chamber at 37 °C. The next day, peptides were eluted by centrifugation at 14,000 g for 20 min, followed by addition of 50 μl of 0.5 M NaCl and further centrifugation. Peptides were desalted using Sep-Pak C18 Plus Light Cartridges (Waters, Milford, MA, USA) or C18 spin columns (Thermo Scientific Pierce, Rockford, IL, USA) according to the manufacturer's instructions. Approximate peptide concentrations were determined using the Pierce Micro BCA assay (Thermo Scientific Pierce, Rockford, IL, USA) following the manufacturer's instructions.

**1D-LC-MS/MS and 2D-LC-MS/MS.** The four biological replicates of each mock community type were analyzed using a block-randomized design as outlined by Oberg and Vitek[51] using several LC-MS/MS methods. Two wash runs with 100% eluent B (80% acetonitrile, 0.1% formic acid) and one blank run were done between samples to reduce carry over. For the 1D-LC-MS/MS mock community runs, 2 μg of peptide were loaded onto a 5 mm, 300 μm ID C18 Acclaim® PepMap 100 pre-column (Thermo Fisher Scientific) using an UltiMate™ 3000 RSLCnano Liquid Chromatograph (Thermo Fisher Scientific) with loading solvent A (2% acetonitrile, 0.05% TFA), eluent A (0.1% formic acid in water) and eluent B. After loading, the pre-column was switched in line with a 50 cm × 75 μm analytical EASY-Spray column packed with PepMap RSLC C18, 2 μm material (Thermo Fisher Scientific), which was heated to 45 °C. The analytical column was connected via an Easy-Spray source to a Q Exactive Plus hybrid quadrupole-Orbitrap mass spectrometer (Thermo Fisher Scientific). Peptides were separated on the analytical column at a flow rate of 225 nl per min and mass spectra acquired in the Orbitrap as described by Petersen et al. (2016)[52]. A 260 min (from 2% B to 31% B in 200 min, in 40 min up to 50% B, 20 min at 99% B) and a 460 min gradient (from 2% B to 31% B in 363 min, in 70 min up to 50% B, 27 min at 99% B) were used for 1D-LC. For the 2D-LC-MS/MS runs, 11 μg of peptide were loaded onto a 10 cm, 300 μm ID Poros 10 S SCX column (Thermo Fisher Scientific) using the UltiMate™ 3000 RSLCnano LC with loading solvent B (2% acetonitrile, 0.5% formic acid). Peptides were eluted from the SCX column onto the C18 pre-column using 20 μl injection of salt plugs from the autosampler with increasing concentrations (12 salt plugs, 0–2000 mM NaCl). After each salt plug injection, the pre-column was switched in line with the 50 cm × 75 μm analytical EASY-Spray column and peptides separated using a 120 min gradient (from 2% B to 31% B in 82 min, in 10 min up to 50% B, 9 min at 99% B, 19 min at 2% B). Data acquisition in the Q Exactive Plus was done as described by Petersen et al.[52].

The two soda lake samples were analyzed in technical quadruplicates by 1D-LC-MS/MS (1 × 260 min and 3 × 460 min runs for each). Two blank runs were done between samples to reduce carry over. For each 260 min run ~1 μg of peptide and for each 460 min run 2–4 μg of peptide were loaded onto a 2 cm, 75 μm ID C18 Acclaim® PepMap 100 pre-column (Thermo Fisher Scientific) using an EASY-nLC 1000 Liquid Chromatograph (Thermo Fisher Scientific) with eluent A (0.2% formic acid, 5% acetonitrile) and eluent B (0.2% formic acid in acetonitrile). The pre-column was connected to a 50 cm × 75 μm analytical EASY-Spray column packed with PepMap RSLC C18, 2 μm material (Thermo Fisher Scientific), which was heated to 35 °C via the integrated heating module. The analytical column was connected via an Easy-Spray source to a Q Exactive Plus. Peptides were separated on the analytical column at a flow rate of 225 nl per min using either a 260 min (from 0 to 20% B in 200 min, in 40 min to 35% B, ending with 20 min at 100% B) or a 460 min gradient (from 0 to 20% B in 354 min, in 71 min to 35% B, ending with 35 min at 100% B). Eluting peptides were ionized with electrospray ionization and analyzed in the Q Exactive Plus as described by Petersen et al.[52].

**Protein identification.** For protein identification of the mock community samples, a database was created using all protein sequences from the reference genomes of the organisms used in the mock communities (Supplementary Table 8). The cRAP protein sequence database (http://www.thegpm.org/crap/) containing protein sequences of common laboratory contaminants was appended to the database. The final database contained 123,100 protein sequences and is available from the PRIDE repository (PXD006118). For protein identification of the soda lake mats we used the database described above. For protein identification of the human saliva metaproteomes we used the same public databases as described in Grassl et al.[9] as a starting point. Namely the protein sequences from the human oral microbiome database[53] and the human reference protein sequences from Uniprot (UP000005640). CD-HIT was used to remove redundant sequences from the database using an identity threshold of 95%[49]. The saliva metaproteome database contained 914,388 protein sequences and is available from the PRIDE repository (PXD006366). For peptide identification and protein inference the MS/MS spectra were searched against the databases using the Sequest HT node in Proteome Discoverer version 2.0.0.802 (Thermo Fisher Scientific) or the MaxQuant software version 1.5.5.1[15].

**Data availability.** The mass spectrometry metaproteomics data and protein sequence databases have been deposited to the ProteomeXchange Consortium via the PRIDE[54] partner repository with the data set identifiers PXD006118 (pure culture and mock community data), PXD006343 (soda lake biomats), and PXD006366 (re-analyses of the saliva metaproteomes by Grassl et al.[9]). A detailed overview of the pure culture and mock community metaproteomic data for method development can be found in Supplementary Table 5.

The sequencing data for the mock community metagenomes and 16S rRNA gene amplicons is available from the European Nucleotide Archive with study accession number PRJEB19901. The 16S rRNA gene amplicon sequencing data of the soda lake biomats have been submitted to the NCBI short read archive (SRA) with the following accession numbers SRR5291562 (Lake1) and SRR5291553 (Lake2). Any other relevant data supporting the findings of the study are available in this article and its Supplementary Information files, or from the corresponding author upon request.

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

## Acknowledgements

We are grateful to Jianwei Chen, Emmo Hamann, Marc Mussmann, Jessica Kozlowski, Lisa Stein, Sean Booth, Jessica Duong, Johanna Voordouw, Kenneth Sanderson, JoongJae Kim, Joenel Alcantara, Anupama P. Halmillawewa, Michael F. Hynes, and Heidi Gibson for donations of cultures for the mock communities, Tjorven Hinzke for providing photographs for the workflow figure, the University of Calgary DNA services core for NextSeq sequencing, Paul Gordon for discussion of NextSeq data analyses, Emil Ruff for discussions on data visualization, Eric Bedford for discussion of statistical analyses, and Niklas Grassl for help with and discussion of the saliva metaproteome data. We thank Leonard Foster and two anonymous reviewers of this paper for their insightful comments and feedback. This study was supported by the Campus Alberta Innovation Chair Program (M.S., C.E.S., X.D., D.L.), the Canadian Foundation for Innovation (M.S.), and the Natural Sciences and Engineering Research Council (NSERC) of Canada through a Banting fellowship (M.K.), an NSERC Undergraduate Student Research Award (E.T.) and a NSERC Discovery Grant to M.S. We acknowledge the support of the Western Canadian Microbiome Center. This research was undertaken thanks in part to funding from the Canada First Research Excellence Fund.

## Author contributions

M.K. conceived study, obtained and created bacterial stocks for mock communities, performed mock community experiments and mass spectrometry, data analysis, wrote the paper with input from all co-authors; E.T. performed the mock community experiments and data analysis, wrote parts of the methods and revised the manuscript; C. E.S. obtained and created bacterial stocks for mock communities, soda lake sampling and sequencing data generation, wrote parts of the methods and revised manuscript; D.L.

proteomics support and experiments, DNA extractions and 16S rRNA gene amplicon library preparation; X.D. assembled and annotated soda lake metagenomes, developed MetaAmp software; C.L. 16S rRNA amplicon library preparation and sequencing on MiSeq; M.S. conceived study, revised manuscript.

## Additional information

**Competing interests:** The authors declare no competing financial interests.

