## [Peer Review File · Nature Communications]

Reviewers' comments:

Reviewer #1 (Remarks to the Author):

This manuscript describes a proteomic approach to quantifying relative contributions to total biomass from component species in microbial communities. It is very comprehensive and overall a very solid study. I have two relatively minor criticisms:

1. Figures 2 and 3 are ridiculously complicated. Fig. 2 might be aided by an additional panel that explains, pictorially, how each of the different filtering systems works. For Fig. 3, in my own mind, putting "Protein Input" at the top of each, rather than at the bottom, might help clarify that the greyed portion of each graph is the 'real' amount.
2. I think that the protein inference problem is still not quite solved here. As far as I can tell, the authors have used the standard inference algorithms in PD, MQ, etc but these don't consider that two proteins IDed by a common peptide may have come from different species. It seems that there is still more to gain in resolving these shared peptides without losing all the sensitivity that is depicted in Fig. 2 as they go to more stringent filtering.

Reviewer: Leonard Foster

Reviewer #2 (Remarks to the Author):

Minor criticism:

1. In the introduction the authors state that the activity is often more important than cell number. That is certainly right, but the authors should comment on the existing stable isotope probing methods and single cell based methods that do exactly that.
2. The argument about the different size and protein content of *S. cerevisiae* and a small bacterium is only correct if protein content equals activity. Which is not necessarily the case.
3. With stating "there is no....." the authors again disregard completely the existence of stable isotope probing methods, including the protein-SIP approach.
4. And although the von Bergen group has established a lot of specific applications of protein-SIP they have missed to evaluate their data in respect to biomass production and turnover.
5. The inference problem is more or less solved by the prophane software and also by the approach used in this study.
6. The authors should comment on the protein content in different growth phases.
7. The question about the validity of genus detection by metaproteomics especially in comparison to community structure assessment by 16srRNA is completely unsolved. The authors underestimate the relevance of this problem (at least for all people working in metaproteomics) and also underestimate the impact their experiments have for answering this question.
8. How do the authors discriminate between peptides and proteins? Which number of peptides have to be detected in order to sum it up to a protein?
9. The workflow shows in the upper part too little detail.
10. The artificial communities are an excellent way of testing the method.
11. Were the communities cultivated or extracted after mixing?
12. Did the authors determine the lower limit of detection (% of proteins existent from one species on the community) ?
13. Did the authors compare this lower limit of detection with the detection by 16srRNA?
14. Why did detection by amplicon sequencing failed so often?
15. The authors should define boundary of acceptable deviation for their analyses in term of cell number detected by different methods.
16. There are many papers out on the inherent problem of 16srRNA amplification, the authors

should mention some of them in the introduction and the discussion part.

17. There is a clear effect of abundance on the accuracy of the detection by metaproteomics, how would this develop with larger communities in which the overall percentage of more species fall under a certain range?

18. The authors should give an example of the development of protein as part of the biomass under different cultivation conditions.

Major criticism:

1. The authors focus on biomass and disregard the approaches that are existent for linking DNA, RNA and proteins to activity.

2. The authors should consider about assessing the protein contribution to the total biomass of a given.

Conclusion:

The study tackles the question of accurate determination of cell and biomass contribution of parts of microbial communities and compare metagenomic approaches and metaproteomics. This in itself is a real achievement. There are aspects missing in the text (link to activity and stable isotope probing approaches) as well as the the question how the methods perform in consortia with higher complexity.

I would recommend a major revision.

Reviewer #3 (Remarks to the Author):

This manuscript describes experimental and analytical methods for quantifying microbial biomass abundance within a community based on proteomic mass spectrometry data. While metaproteomic methods have been used on a variety of communities for assessing protein abundance, most studies have not attempted to aggregate and accurately quantify protein contributions at the species level in a manner analogous to the way 16S amplicon data or metagenome data are used to quantify cellular / genome abundance. In their study the investigators apply their methods to both mixes of known organisms (including eukaryotes, prokaryotes and DNA and RNA viruses) and environmental samples to demonstrate effectiveness.

Overall the study is well done and a valuable contribution to the field, though no new biological findings are presented. The analysis, presentation and statistics used are appropriate.

Specific comments:

Line 284: "All three methods performed badly, when it came to estimating the species cell numbers in the samples..." I'm not sure if there's a better way to phrase it but it seems odd to say they performed "badly" when they've performed more or less as expected but the quantity measured isn't a good proxy for cell numbers.

Line 428: "Since, the cell lysis method used for both approaches was identical an extraction bias is unlikely, suggesting that a primer bias may be responsible for the discrepancy." It seems equally likely that relic DNA from dead organisms could also explain a discrepancy between DNA abundance and protein abundance. Also no comma is needed after "Since".

Lines 472-479 compare metaproteomics and sequencing based methods as though it's an either/or choice, but in fact since metagenome sequencing is generally necessary for creating a reference for proteomics it seems more accurate to think in terms of augmenting sequencing-based methods with metaproteomics.

Minor comments:

Line 365: "...allow to correct the relative..." should be "...allow correction of the relative..." or "...allow one to correct the relative..." Same correction at line 472, "...allows to accurately quantify..."

Line 417: "...we generated both metaproteomic data, as well as 16 rRNA gene amplicon data..." is redundant; can remove "both" or change "as well as" to "and".

Line 491: "Second, a likely much more difficult question to answer is, if and under what circumstances proteinaceous biomass of a community member can be used as an approximation of the biological activity of that community member?" This phrasing is pretty awkward; I'd change to "Second, a likely much more difficult question to answer is, can proteinaceous biomass of a community member be used as an approximation of the biological activity of that community member, and if so under what circumstances?"

Line 535: "steps" should be "step"

Line 566: Is there a reason for leaving out the genus name for *Roseobacter* sp. AK199?

Reviewer #1 Leonard Foster (Remarks to the Author):

This manuscript describes a proteomic approach to quantifying relative contributions to total biomass from component species in microbial communities. It is very comprehensive and overall a very solid study. I have two relatively minor criticisms:

1. Figures 2 and 3 are ridiculously complicated. Fig. 2 might be aided by an additional panel that explains, pictorially, how each of the different filtering systems works. For Fig. 3, in my own mind, putting "Protein Input" at the top of each, rather than at the bottom, might help clarify that the greyed portion of each graph is the 'real' amount.

We agree with the reviewer. For figure 2 we have now simplified the figure by making the text and abbreviations both in the figure and legend much more concise and by highlighting the specificity and sensitivity related graphs with appropriate axis labels.

To simplify figure 3 we decided to split it into two figures (now figure 3 and 4). One for the comparison of the protein quantification methods and one for the comparison of metaproteomics with DNA sequencing based methods. We also included your suggestion to move "protein input" to the top and relabeled it as "actual".

2. I think that the protein inference problem is still not quite solved here. As far as I can tell, the authors have used the standard inference algorithms in PD, MQ, etc but these don't consider that two proteins IDed by a common peptide may have come from different species. It seems that there is still more to gain in resolving these shared peptides without losing all the sensitivity that is depicted in Fig. 2 as they go to more stringent filtering.

We agree with the reviewer that the protein inference problem per se is not solved in our study. By definition the inference problem cannot be “solved” entirely for bottom-up proteomics, because when a peptide occurs in multiple proteins, it cannot be assigned to either. However, in Fig 2 we show that by applying strict criteria for identification of proteins, mis-assignment of proteins to other species can be mostly avoided. This comes at the expense of some loss of sensitivity. Therefore, we provide an assessment of the consequences of the inference at different taxonomic levels and provide identification strategies that enable accurate determination of species biomass contributions (see also Fig 3 and Fig 4), which was the aim of our study. The extensive dataset that we provide with our study will enable future tweaking of inference parameters when new approaches become available.

Reviewer #2 (Remarks to the Author):

Minor criticism:

1. In the introduction the authors state that the activity is often more important than cell number. That is certainly right, but the authors should comment on the existing stable isotope probing methods and single cell based methods that do exactly that.

Thank you for bringing up this point, it made us realize that our use of the word “activity” can lead to some confusion about the scope of the paper, which really is assessment of proteinaceous biomass and not measurement of activity. To avoid confusion we have now removed the word activity from the two sentences in the introduction that provided cause for confusion.

These sentences now read:

“Cell numbers, however, are often not the best measure for a species’ contribution to a community, because microbial cells can differ by several orders of magnitude in biomass.”

“Currently, there are no high-throughput methods available to estimate the biomass contribution of individual community members.”

We included one sentence in the discussion, which poses the question if proteinaceous biomass could potentially be a measure for total biological activity of a species, but we leave this question deliberately open as the answer to that may require many future studies. This sentence in the discussion reads:

“Second, a likely much more difficult question to answer is, can proteinaceous biomass of a community member be used as an approximation of the biological activity of that community member, and if so under what circumstances?”

2. The argument about the different size and protein content of *S. cerevisiae* and a small bacterium is only correct if protein content equals activity. Which is not necessarily the case.

Please see our response to your comment 1.

3. With stating “there is no.....” the authors again disregard completely the existence of stable isotope probing methods, including the protein-SIP approach.

Please see our response to your comment 1.

4. And although the von Bergen group has established a lot of specific applications of protein-SIP they have missed to evaluate their data in respect to biomass production and turnover.

We agree that there is certainly a lot of room for developing proteomics based methods, which ideally will provide measures of total biomass production and turnover.

Please also see our response to your comment 1.

5. The inference problem is more or less solved by the prophane software and also by the approach used in this study.

Thank you for pointing us to the Prophane software, we did not know this tool beforehand. We have now added a citation to Prophane and to The MetaProteomeAnalyzer pipeline in the discussion section. The respective sentence reads:

“To make the approach as broadly applicable as possible, we have chosen to use a route that should make it possible to transfer the approach to many other platforms including some recently developed software pipelines addressing the protein inference problem in metaproteomics such as The MetaProteomeAnalyzer¹ and Prophane².”

We agree that for the purpose of protein taxonomy inference down to the species level the inference problem is close to being solved, however, the general protein inference problem can likely never be entirely solved for bottom-up proteomics.

Please also see our response to comment #2 or reviewer #1.

6. The authors should comment on the protein content in different growth phases.

This is an interesting point but it is outside the scope of our study, because we do not look at activity and protein turnover. To avoid confusing the readers we would thus prefer not to discuss this topic in the paper. Future studies could address this point using the methods presented here.

7. The question about the validity of genus detection by metaproteomics especially in comparison to community structure assessment by 16srRNA is completely unsolved. The authors underestimate the relevance of this problem (at least for all people working in metaproteomics) and also underestimate the impact their experiments have for answering this question.

Thank you. The fact that this problem has not been addressed with controlled datasets in the past is the reason why we did such an extensive study to address the question if and how community structure can be assessed with metaproteomics. We changed the respective heading in the results section to emphasize our findings more. It now reads:

“Metaproteomics enables accurate species-level quantification”

8. How do the authors discriminate between peptides and proteins? Which number of peptides have to be detected in order to sum it up to a protein?

We are not sure what the reviewer means with “discriminating between peptides and proteins”. The search engines (SEQUEST in Proteome Discoverer and Andromeda in MaxQuant) provide lists of identified peptides based on the matching of MS2 spectra to database derived peptide sequences. The protein inference algorithms then matches the peptides to protein sequences and uses statistical models or specific filtering criteria to group proteins or to predict which protein sequence is the actually expressed one. The number of spectra and peptides matching to a protein are provided as an integer value. The protein inference strategy that we used most in the manuscript requires that the protein has to pass through the protein inference algorithm FidoCT at a false discovery rate <5% and in addition to that has at least two protein-unique peptides. Protein-unique peptides refers to those peptide sequences that only match a single protein sequence in the database. This strategy is explained in the result section with the heading “Achieving high specificity with minimal losses in sensitivity”.

9. The workflow shows in the upper part too little detail.

We kept the upper part of the workflow figure pretty generic on purpose, because in our study we did not test the influence of sample processing and LC-MS/MS approaches on community structure assessment. By keeping it generic we try to avoid the impression that we recommend a very specific sample and LC-MS/MS protocol. We highlighted the important steps for the data processing and analyses in great detail. The specific sample processing and LC-MS/MS approaches are described in great detail in the methods.

10. The artificial communities are an excellent way of testing the method.

We thank the reviewer for this supportive statement.

11. Were the communities cultivated or extracted after mixing?

After mixing and aliquoting the communities the aliquots were frozen at -80°C until they were extracted. We have now clarified this in the methods section, which now reads:

“... Four biological replicates of each mock community type were made and each replicate was divided into 20 aliquots. The aliquots were frozen at -80°C until extraction. The UNEVEN mock community was designed...”

12. Did the authors determine the lower limit of detection (% of proteins existent from one species on the community)?

Yes we did analyze the detection limit of the method in great detail, this is described in the results section, which is now titled “How much data is needed”. The detection limit depends on how much data is available (i.e. number of MS2 spectra), as well as on the organism type. We found, for example, that with 260 minute LC-MS/MS runs the detection limit for Bacteria is below 0.08% of the total protein, while for bacteriophages the detection limit was much higher.

13. Did the authors compare this lower limit of detection with the detection by 16srRNA?

We could not conceive of a way to do a fair detection limit comparison for metaproteomics and 16S rRNA amplicon sequencing, because the detection limit of both methods very much depends on the amount of data produced and the data types are not comparable. We did, however, do an extensive comparison of detection by metaproteomics and 16S rRNA gene amplicon sequencing, which is described in the section that is now entitled “Metaproteomics is more accurate for biomass estimates than sequencing methods”. We found that metaproteomics has, as expected, a much broader range of detection as compared to amplicon sequencing. See our response to your comment 14.

14. Why did detection by amplicon sequencing failed so often?

Detection failed for numerous reasons. 1) the phages do not have a 16S rRNA gene, 2) the primers were bacteria specific thus the Archaeon and the Green algae were excluded, 3) as always in 16S rRNA sequencing the primers exhibit strong biases favoring some bacterial species over others. We discuss the reasons for detection failure by amplicon sequencing, for example, in the following sentences in the results section and also provide the necessary citations:

“The remaining 17 OTUs were taxonomically classified by MetaAmp at the genus level. A species level classification was not possible because of the limited information content of the amplicon sequences. This meant, for example, that there were three OTUs that were classified as Pseudomonas.”

“As expected none of the bacteriophages were detected by amplicon sequencing due to the absence of a 16S rRNA gene in these phages (Fig. 4a). We also did not detect the Archaeon N. viennensis, the eukaryotic green algae Chl. reinhardtii and six of the bacterial species by amplicon sequencing. The primer pair that we used to generate the amplicons is optimized for the greatest possible coverage of the bacterial domain³, therefore it was not surprising that N. viennensis and Chl. reinhardtii were not detected, although we successfully amplified at least the chloroplast sequence of green algae using this primer pair in the past (data not shown).”

“The failure to detect some of the bacteria in all replicates is harder to explain. We have successfully generated amplicons from pure cultures of N. europaeae, N. ureae and N. multififormis in the past with the primer pair used here (data not shown), thus we have to assume that these species were not detected due to their low abundance in the UNEVEN community samples or due to a primer bias leading to preferential amplification of the other bacterial

species. Such primer biases are a known problem for 16S rRNA gene amplicon sequencing^{4,5}. For the R. leg. bv. viciae and S. aureus strains the amplicon sequences did not distinguish between each of the two strains in the samples and thus only a minimum of one strain detection per species could be corroborated.”

15. The authors should define boundary of acceptable deviation for their analyses in term of cell number detected by different methods.

As the boundary of acceptable deviation very much depends on the research question we do not feel comfortable defining a boundary. However, to highlight the fact that deviation is an issue and that deviation depends on the abundance of species in the sample we did an analysis of the correlation of deviation from the actual and species abundance. This is described in the following section:

“Interestingly, all five approaches underestimated the abundances of species/strains that are present in the samples in low amounts (Fig. 5a). If low-abundance species (<0.5% in all approaches) are removed from the dataset resulting in 18 species remaining, then the deviation of the measurement from the actual protein input amount becomes much smaller (Fig. 5b, Supplementary Table 6). This suggests that, as with most other analytical methods, the accuracy of the measurement is lower for quantities close to the detection limit and thus the proteinaceous biomass estimates for low abundant species should be treated as less precise”

16. There are many papers out on the inherent problem of 16srRNA amplification, the authors should mention some of them in the introduction and the discussion part.

The following two sentences in the manuscript refer to the 16S rRNA amplification bias problem and provide the necessary citations:

“Such primer biases are a known problem for 16S rRNA gene amplicon sequencing^{4,5}”

“Analogous to the primer bias based exclusion or incorrect estimation of species abundances in 16S/18S rRNA gene amplicon sequencing^{3,6}, the incompleteness of the protein sequence database used for protein identification can lead to the exclusion or incorrect estimation of species abundances”

17. There is a clear effect of abundancy on the accuracy of the detection by metaproteomics, how would this develop with larger communities in which the overall percentage of more species fall under a certain range?

Please also see our response to your comment 15. We agree, if the abundance of a species falls below 0.5% the accuracy of quantification decreases for metaproteomics, however, the same is true for the sequencing based methods. Metaproteomics is surprisingly sensitive though (see also our response to your comment 12.) as species down to 0.08% abundance are still detected albeit with less accuracy on the quantification side. Most studies in microbial ecology and environmental microbiology focus on the more dominant species in a community and we feel that

the methods that we present are clearly suited to quantify abundant and less abundant species in communities. Quantification of rare species, however, would not be possible at the moment. This is where also sequencing based methods fail to provide accurate quantification. If quantification of rare species is the aim of the study, methods such as fluorescence in situ hybridization, real time PCR and miro-arrays might be used, if sequence information is available.

18. The authors should give an example of the development of protein as part of the biomass under different cultivation conditions.

This is an interesting point. Thank you for bringing it up. We have now done an extensive literature search regarding this topic. The consensus based on the literature is that the ratio of protein to cell dry weight generally does not change much during different growth stages, however, the protein to cell volume ratio may change quite a bit. Proteinaceous biomass as determined by our approach should thus be a good estimator for total cell biomass. We have now included a sentence in the discussion with three references that points to the relative stability of protein to dry cell mass ratios so that readers are aware of it. It reads:

“We would argue that in many cases, proteinaceous biomass is a good estimate of total biomass, because it has been shown for a variety of bacteria that the ratio of protein to total cell dry weight is relatively constant even for different growth states⁷⁻⁹. However, as always we expect exceptions, where proteinaceous biomass is not a good predictor of total biomass, which would for example be the case of microorganisms that store large amounts of carbon in form of polyhydroxyalkanoates or glycogen.”

Major criticism:

1. The authors focus on biomass and disregard the approaches that are existent for linking DNA, RNA and proteins to activity.

Please see our responses to your comments 1. and 4.

2. The authors should consider about assessing the protein contribution to the total biomass of a given.

Please see our response to your comment 18.

Conclusion:

The study tackles the question of accurate determination of cell and biomass contribution of parts of microbial communities and compare metagenomic approaches and metaproteomics. This in itself is a real achievement. There are aspects missing in the text (link to activity and stable isotope probing approaches) as well as the the question how the methods perform in consortia with higher complexity.

I would recommend a major revision.

Reviewer #3 (Remarks to the Author):

This manuscript describes experimental and analytical methods for quantifying microbial biomass abundance within a community based on proteomic mass spectrometry data. While

metaproteomic methods have been used on a variety of communities for assessing protein abundance, most studies have not attempted to aggregate and accurately quantify protein contributions at the species level in a manner analogous to the way 16S amplicon data or metagenome data are used to quantify cellular / genome abundance. In their study the investigators apply their methods to both mixes of known organisms (including eukaryotes, prokaryotes and DNA and RNA viruses) and environmental samples to demonstrate effectiveness.

Overall the study is well done and a valuable contribution to the field, though no new biological findings are presented. The analysis, presentation and statistics used are appropriate. Specific comments:

Line 284: “All three methods performed badly, when it came to estimating the species cell numbers in the samples...” I’m not sure if there’s a better way to phrase it but it seems odd to say they performed “badly” when they’ve performed more or less as expected but the quantity measured isn’t a good proxy for cell numbers.

We agree that this was phrased “badly” and have therefore rewritten the sentence. It now reads:

“The relative species abundances provided by all three methods did not correlate well with the actual cell numbers in the samples (Fig. 4d).”

Line 428: “Since, the cell lysis method used for both approaches was identical an extraction bias is unlikely, suggesting that a primer bias may be responsible for the discrepancy.” It seems equally likely that relic DNA from dead organisms could also explain a discrepancy between DNA abundance and protein abundance. Also no comma is needed after “Since”.

Good point, thank you. We rewrote the sentence and added the necessary citations for relic DNA (Carini et al. and Lennon et al). It now reads:

“Since the cell lysis method used for both approaches was identical an extraction bias is unlikely, suggesting that other causes such as a primer bias or relic DNA^{10, 11} may be responsible for the discrepancy.”

Lines 472-479 compare metaproteomics and sequencing based methods as though it’s an either/or choice, but in fact since metagenome sequencing is generally necessary for creating a reference for proteomics it seems more accurate to think in terms of augmenting sequencing-based methods with metaproteomics.

We agree that this point was not completely clear. We have now rewritten this paragraph to clarify. It now reads:

“As we demonstrate here, metaproteomics-based biomass assessment is a powerful approach that allows accurate quantification of the proteinaceous biomass of a large number of taxa in a community all at once. This approach augments existing high throughput approaches for determining community structure based on DNA sequencing, in that it provides an additional,

independent measure of community structure. Our case study on soda lake biomass nicely illustrates that sequencing-based methods and metaproteomics can provide very different pictures of a community. An added benefit of using metaproteomes in addition to sequencing based methods for community structure analyses is that the proteomic information will also provide insights into which metabolic and physiological functions are expressed and play a major role in the community.”

Minor comments:

Line 365: “...allow to correct the relative...” should be “...allow correction of the relative...” or “...allow one to correct the relative...” Same correction at line 472, “...allows to accurately quantify...”

Done, thank you.

Line 417: “...we generated both metaproteomic data, as well as 16 rRNA gene amplicon data...” is redundant; can remove “both” or change “as well as” to “and”.

Done

Line 491: “Second, a likely much more difficult question to answer is, if and under what circumstances proteinaceous biomass of a community member can be used as an approximation of the biological activity of that community member?” This phrasing is pretty awkward; I’d change to “Second, a likely much more difficult question to answer is, can proteinaceous biomass of a community member be used as an approximation of the biological activity of that community member, and if so under what circumstances?”

Done

Line 535: “steps” should be “step”

Done

Line 566: Is there a reason for leaving out the genus name for *Roseobacter* sp. AK199?

We now added the genus name.

1. Muth, T. et al. *The MetaProteomeAnalyzer: a powerful open-source software suite for metaproteomics data analysis and interpretation. J. Proteome Res.* **14**, 1557-1565 (2015).
2. Schneider, T. et al. *Structure and function of the symbiosis partners of the lung lichen (*Lobaria pulmonaria* L. Hoffm.) analyzed by metaproteomics. Proteomics* **11**, 2752-2756 (2011).
3. Klindworth, A. et al. *Evaluation of general 16S ribosomal RNA gene PCR primers for classical and next-generation sequencing-based diversity studies. Nucleic Acids Res.* **41**, e1 (2013).

4. Zhou, J. et al. High-throughput metagenomic technologies for complex microbial community analysis: open and closed formats. *mBio* **6**, e02288-02214 doi: 10.1128/mBio.02288-14 (2015).
5. Parada, A.E., Needham, D.M. & Fuhrman, J.A. Every base matters: assessing small subunit rRNA primers for marine microbiomes with mock communities, time series and global field samples. *Environ. Microbiol.* **18**, 1403-1414 (2016).
6. Tremblay, J. et al. Primer and platform effects on 16S rRNA tag sequencing. *Frontiers in Microbiology* **6**, 771 (2015).
7. Simon, M. & Azam, F. Protein content and protein synthesis rates of planktonic marine bacteria. *Mar. Ecol. Prog. Ser.* **51**, 201-213 (1989).
8. Brunschede, H., Dove, T.L. & Bremer, H. Establishment of exponential growth after a nutritional shift-up in *Escherichia coli* B/r: accumulation of deoxyribonucleic acid, ribonucleic acid, and protein. *J. Bacteriol.* **129**, 1020-1033 (1977).
9. Shahab, N., Flett, F., Oliver, S.G. & Butler, P.R. Growth rate control of protein and nucleic acid content in *Streptomyces coelicolor* A3(2) and *Escherichia coli* B/r. *Microbiology* **142**, 1927-1935 (1996).
10. Carini, P. et al. Relic DNA is abundant in soil and obscures estimates of soil microbial diversity. *Nature Microbiology* **2**, 16242 doi: 10.1038/nmicrobiol.2016.242 (2016).
11. Lennon, J.T., Placella, S.A. & Muscarella, M.E. Relic DNA contributes minimally to estimates of microbial diversity. *bioRxiv*, 131284 doi: 10.1101/131284 (2017).

REVIEWERS' COMMENTS:

Reviewer #2 (Remarks to the Author):

Dear authors,

congratulation to this sound study.

I was satisfied with almost of your answers to my questions and remarks.

Wisely you exchanged the activity terms against biomass, which fits much better to the design of the study. I am still convinced that for many of the colleagues working in the field of metaproteomics it would be helpful to clarify this difference expressively in the manuscript and mention the ways how activity can be linked to protein synthesis.

My recommendation: Accept with (very) minor revision

Reviewer #2 (Remarks to the Author):

Dear authors,

Congratulation to this sound study.

I was satisfied with almost of your answers to my questions and remarks.

Wisely you exchanged the activity terms against biomass, which fits much better to the design of the study. I am still convinced that for many of the colleagues working in the field of metaproteomics it would be helpful to clarify this difference expressively in the manuscript and mention the ways how activity can be linked to protein synthesis.

Dear Reviewer #2,

Thank you for your congratulations. We very much appreciate your constructive feedback on the manuscript.

We have now included one sentence at the end of the discussion that highlights methods for measuring species specific activities in microbial communities. The last paragraph of the discussion now reads:

“There are several questions that go beyond the scope of this study that should be addressed in the future. First, is proteinaceous biomass an accurate representation of the total biomass of a species? We would argue that in many cases, proteinaceous biomass is a good estimate of total biomass, because it has been shown for a variety of bacteria that the ratio of protein to total cell dry weight is relatively constant even for different growth states²⁸⁻³⁰. However, as always we expect exceptions, where proteinaceous biomass is not a good predictor of total biomass, which would for example be the case of microorganisms that store large amounts of carbon in form of polyhydroxyalkanoates or glycogen. Second, a likely much more difficult question to answer is, can proteinaceous biomass of a community member be used as an approximation of the biological activity of that community member, and if so under what circumstances? This question can potentially be addressed in the future by combining the metaproteomics-based biomass assessment approach with methods that allow determination of species-specific activities based on incorporation of stable isotopes on the single-cell level such as NanoSIMS³¹ and Raman microspectroscopy³² or community-level by metaproteomics using Protein-SIP³³.”

31. Musat, N., Musat, F., Weber, P.K. & Pett-Ridge, J. Tracking microbial interactions with NanoSIMS. *Curr. Opin. Biotechnol.* **41**, 114-121 (2016).
32. Wagner, M. Single-cell ecophysiology of microbes as revealed by Raman microspectroscopy or secondary ion mass spectrometry imaging. *Annu. Rev. Microbiol.* **63**, 411-429 (2009).
33. Jehmlich, N., Vogt, C., Lünsmann, V., Richnow, H.H. & von Bergen, M. Protein-SIP in environmental studies. *Curr. Opin. Biotechnol.* **41**, 26-33 (2016).